# Stem Cell Therapy for Aging Related Diseases and Joint Diseases in Companion Animals

**DOI:** 10.3390/ani13152457

**Published:** 2023-07-29

**Authors:** Yanmin Wang, Michael Alexander, Todd Scott, Desiree C. T. Cox, Augusta Wellington, Mike K. S. Chan, Michelle B. F. Wong, Orn Adalsteinsson, Jonathan R. T. Lakey

**Affiliations:** 1California Medical Innovations Institute, 11107 Roselle Street, San Diego, CA 92121, USA; 2Department of Surgery, University of California Irvine, Irvine, CA 92868, USA; 3Crestwood Veterinary Clinic, Edmonton, AB T5P 1J9, Canada; 4European Wellness Group, Klosterstrasse 205ID, 67480 Edenkoben, Germany; 5Graduate Faculty, School of Graduate Studies, Rutgers University, New Brunswick, NJ 07013, USA; 6Department of Biomedical Engineering, University of California Irvine, Irvine, CA 92697, USA

**Keywords:** stem cell therapy, mesenchymal stem cells, regenerative medicine, aging, veterinary use, companion animals

## Abstract

**Simple Summary:**

Companion animals suffer from ailments that are not curable by conventional veterinary medicine. Stem cells offer a new treatment avenue for these ailments because of their regenerative potential resulting in local tissue regeneration and repair. This review will summarize the various uses of stem cells as treatment for age-related and joint diseases in companion animals.

**Abstract:**

Stem cell therapy is an attractive treatment for diseases in companion animals that cannot be treated by conventional veterinary medicine practices. The unique properties of stem cells, particularly the ability to differentiate into specific cell types, makes them a focal point in regenerative medicine treatments. Stem cell transplantation, especially using mesenchymal stem cells, has been proposed as a means to treat a wide range of injuries and ailments, resulting in tissue regeneration or repair. This review aims to summarize the veterinary use of stem cells for treating age-related and joint diseases, which are common conditions in pets. While additional research is necessary and certain limitations exist, the potential of stem cell therapy for companion animals is immense.

## 1. Introduction

Stem cell therapies have been widely studied since the initial use of bone marrow transplantation for hematologic diseases in the 1950s [1]. Since then, interest and research into further applications of stem cells, particularly in regenerative medicine, have greatly expanded [2,3]. At their core, stem cells are undifferentiated cells capable of self-renewal, regeneration, and differentiation into specific cell types, allowing the regeneration and repair of damaged tissue [4,5]. Stem cells also possess the capacity to generate peptides that hold potential for therapeutic applications in the treatment of various diseases. For example, the clinical outcomes of organ-specific mammalian precursor stem cell-derived peptide preparations [6,7,8], manufactured by European Wellness laboratories, have been documented in both historical and scientific journals [9,10,11] across a wide range of diseases and illnesses where treatment options are limited.

In recent years, stem cell therapy has gained popularity and generated significant interest in both research and clinical settings. In veterinary medicine, stem cell therapy has been explored as a potential treatment for a wide range of health issues, including dermatologic, dental, endocrine, neurological, cardiovascular, respiratory, urinary, and gastrointestinal diseases [12,13,14,15]. The use of stem cells for these conditions is still in the early stages of research, but the potential for stem cell therapy in the field of veterinary medicine is vast.

This review focuses specifically on the clinical application of stem cells in the treatment of age-associated diseases and joint diseases in pets, providing an overview of the current state of research and potential future directions for stem cell therapy in veterinary medicine. To our knowledge, it is the first review paper on this topic. It highlights recent findings on the use of stem cells in treating orthopedic diseases such as osteoarthritis, hip dysplasia, and cranial cruciate ligament rupture, as well as the potential benefits of stem cell therapy in aging pets. Additionally, it explores the challenges and limitations of the use of stem cells in veterinary medicine. Ultimately, the goal of this review paper is to provide a comprehensive understanding of the current state of stem cell therapy in veterinary medicine and to explore the potential for this innovative treatment to revolutionize the management of age-related disorders and joint diseases in companion animals.

## 2. Type of Stem Cells

There are two main categories of stem cells, which are pluripotent stem cells (embryonic stem cells and induced pluripotent stem cells) and nonembryonic or somatic stem cells (commonly called “adult” stem cells) [5]. Embryonic stem cells can differentiate into all of the cells of the adult body; however, these stem cells have been a topic of ethical controversy due to the destruction of human embryos required to obtain these cells [16]. As a result, the use of embryonic stem cells is heavily restricted. In contrast, induced pluripotent stem cells can be generated by reprogramming adult cells and therefore do not raise the same ethical concerns [16]. However, there are limitations to their use as they may retain some characteristics of the original cell type, potentially leading to genetic mutation, tumor formation, or other unwanted effects [17].

Adult stem cells can be derived from various body tissues, such as bone marrow, peripheral blood, umbilical cord blood and tissue, adipose tissue, skin, neurons, and muscle [18]. Among adult stem cells, mesenchymal stem cells (MSCs) are of particular clinical relevance. MSCs originate from the connective tissue or stroma surrounding organs and other tissues. They are easy to isolate from various tissues such as bone marrow, adipose tissue, or umbilical cord tissue using well-established protocols that add to their ease of use and clinical superiority [19,20]. After collection, tissues are processed to isolate the desired cell population. The isolated cells are then cultured and expanded in laboratory conditions to increase their numbers. This expansion phase allows for the generation of a larger quantity of stem cells for therapeutic use. The common locations that MSCs are isolated from in companion animals are bone marrow or adipose tissue; although site-specific differences have been observed regarding stem cell numbers and differentiation capacity [19,21]. Briefly, bone marrow and adipose tissue offer a larger number of MSCs than other tissues. In addition, canine adipose tissue derived MSCs were reported to have higher proliferative potential, whereas bone marrow derived MSCs showed a higher secretory production of soluble factors and exosomes [19,22]. These differences are important to note when optimizing the design and execution of clinical trials. Extensive work towards the characterization of MSCs based on cell-surface markers and multipotency has been performed [20]. MSCs have the ability to differentiate into various cell lineages and also possess immunological properties such as anti-inflammatory, immunoregulatory, and immunosuppressive capacities [20,23]. These properties enhance their potential for use as immune-tolerant agents. As a result, MSCs have demonstrated efficacy in the treatment of various conditions, making them a promising option for regenerative medicine [19,20].

Compared to products with a mixture of cells, such as stromal vascular fraction [24] and bone marrow aspirate concentrate (BMAC) [25], MSCs and other pure stem cell products have a more standardized cell population, a higher stem cell concentration, and a lower risk of immune reaction associated with non-stem cell components. Stem cells can also be utilized in conjunction with other products such as hyaluronic acid [26] and platelet-rich plasma (PRP) [27,28], which have shown favorable outcomes in treatment.

Typically, there are two approaches to stem cell transplantation: autologous infusion, where stem cells are generated from the patient being treated, and allogeneic infusion, where stem cells are generated from a genetically distinct individual [29]. Autologous stem cell infusion has been widely accepted due to its perceived advantages in preventing or treating early intragraft inflammation and reducing the risk of acute rejection [29]. On the other hand, the unique properties of stem cells, such as their low immunogenicity and immunomodulatory capabilities, make allogeneic use possible [21,23]. Allogeneic infusion holds promise for utilizing cells from healthy donors with robust regenerative capabilities. Both autologous and allogeneic infusion have been reported in the field of veterinary medicine.

## 3. Stem Cells for Age-Related Diseases

Aging is a natural and inevitable process that affects all living organisms including pets. As pets age, they become increasingly susceptible to various age-related diseases, such as cognitive dysfunction syndrome (CDS) [30,31], cardiac dysfunction [32], and other disorders. These diseases can significantly impact the quality of life of pets. Cellular senescence, characterized by the irreversible loss of proliferative potential in somatic cells, plays a crucial role in age-related tissue degeneration [33,34]. This fundamental understanding provides a rationale for the potential of stem cells in mitigating age-related disorders.

CDS is a common condition in senior dogs and cats, which is characterized by a progressive decline in cognitive function, memory loss, disorientation, and behavior changes, similar to dementia or Alzheimer’s disease in humans [35,36]. CDS is believed to be caused by multiple factors, including oxidative stress, inflammation, and changes in neurotransmitter function [37,38,39]. While there is no cure for CDS [35], stem cell therapy has shown promise in treating this disease in human and animal models [40,41], as stem cells have the ability to differentiate into neurons and can promote neurogenesis. However, there has been limited application in veterinary CDS therapy. Valenzuela et al. [42] injected autologous skin-derived neural precursor cells into the bilateral hippocampus of dogs with a definitive diagnosis of CDS and observed improvements or complete reversal of brain function, along with neurosynaptic restoration in the hippocampus [42]. This suggests that stem cell therapy may be a potential treatment option for dogs with CDS. Further research is needed to explore the efficacy and feasibility of stem cell therapy for CDS in pets.

Myxomatous mitral valvular disease (MMVD), also known as degenerative mitral valve disease, is the most common naturally acquired heart disease and the leading cause of congestive heart failure (CHF) in dogs [43]. MMVD is an age-related disease, as the degenerative changes in the valve associated with MMVD tend to occur gradually over time. A review [44] indicated that the incidence of MMVD over a canine lifetime is almost 100%. Stem cell therapy has been applied in the treatment of MMVD and CHF, but the outcomes in dogs are not clear. Yang et al. [43] conducted a double-blind, placebo-controlled trial in dogs with CHF secondary to MMVD and indicated that intravenous infusion of allogeneic Wharton jelly (umbilical tissue)-derived MSCs was safe; however, no significant therapeutic effects were observed in terms of echocardiographic data, electrocardiogram results, serum cardiac biomarker concentrations, and survival time when compared to autologous serum therapy [43]. On the contrary, Petchdee and Sompeewong [45] found that the intravenous administration of puppy deciduous teeth stem cells resulted in improvements in left ventricular ejection fraction, heart function, and quality of life in elderly dogs with MMVD. We hypothesize that the discrepancy in outcomes may result from the different sources or types of cells used and that larger sample sizes are required for further evaluation.

Chronic kidney disease (CKD) is a prevalent age-related condition that causes significant morbidity and mortality in pets, particularly in cats [46], by causing gradual damage and deterioration of the kidneys. Presently, there is no definitive cure for CKD apart from renal transplantation. Studies have evaluated the safety and efficacy of MSCs in treating this disease. In a phase I clinical trial, Thomson et al. [47] delivered autologous adipose tissue-derived MSCs intra-arterially to the kidney in cats with CKD over a period of 3 months and demonstrated the safety and feasibility of this treatment. Quimby et al. [48] applied unilateral intrarenal injection of autologous adipose tissue or bone marrow-derived MSCs to cats with stable CKD and observed no short- or long-term adverse events. In some animals, the injection induced a slight increase in glomerular filtration rate and a mild decrease in serum creatinine concentration. However, in larger-size pilot studies and a randomized controlled trial, the use of allogeneic MSCs did not result in improved renal function parameters [49,50]. This discrepancy in treatment outcomes does not appear to be associated with the source of the stem cells (autologous vs. allogeneic), as other studies have reported a beneficial effect of allogeneic MSCs therapy in CKD. For example, Vidane et al. [51] found that in cats with naturally occurring CKD, the intravenous infusion of allogeneic MSCs derived from the amniotic membrane improved renal function by decreasing urine protein-to-creatinine ratio and serum creatinine concentration, and led to increased food intake and social behavior. Additional clinical trials are required to demonstrate the efficacy of stem cells in CKD treatment.

There are several other age-related diseases in pets, including insulin-independent diabetes, periodontal disease, cataracts, and cancer, for which the use of stem cell therapy is currently not widely practiced in the veterinary industry and for which no clinical studies are available.

## 4. Stem Cells for Joint Diseases

Joint diseases such as osteoarthritis (OA) and ligament ruptures are common conditions in pets [52,53]. Traditional treatments for joint diseases include surgery and pain management, but these treatments are not always effective and can have side effects [54]. Stem cells, particularly MSCs derived from adipose tissues or bone marrow, have been shown to differentiate into cartilage cells, promote the growth of new cartilage, and confer anti-inflammatory properties [54,55,56]. As a result, they are now widely used in the treatment of orthopedic injuries and joint diseases in pets [57], making stem cell therapy a promising option in this field.

Dogs are susceptible to the development of orthopedic injuries or arthritis due to various factors, including congenital conditions, trauma, genetics, and other predispositions such as obesity, old age, and overuse. Around one in four of the 77.2 million pet dogs in the US are diagnosed with some form of arthritis, with over 20% of them being affected by OA once they are over the age of one [52]. OA, also known as “Degenerative Joint Disease”, is the most prevalent form of arthritis in dogs, and pain is the primary symptom observed [52]. Research indicates that OA is not a normal consequence of aging or cartilage degeneration in dogs, despite its frequent diagnosis in elderly canines [58]. Rather, the primary risk factor of OA is weight or body condition score [59,60], and it is often associated with trauma such as abnormal loading on a normal joint or normal force on an abnormal joint [61].

Suspicion of and subsequent diagnosis of OA is more difficult in cats than in dogs, as pain secondary to OA is more easily interpreted in dogs [62], and cats tend to use behavioral strategies to disguise pain and lameness [63,64]. A study on 100 cats aged 6–19 years indicated that over 60% of cats in the sample had OA in at least one joint [64]. In cats, OA appears to be a primary disease rather than secondary to other diseases as is commonly observed in dogs [64].

The main feature of OA [65] is cartilage damage or loss, and the poor ability of cartilage lesions to heal makes it challenging to develop effective therapies. In the past, traditional treatments for OA, such as analgesics and anti-inflammatory drugs, have been used to manage symptoms, but these do not address the underlying cause of cartilage damage and/or promote healing. In recent years, there have been several reports using stem cells for the treatment of OA in pets. Shah et al. [66] studied over 200 dogs suffering from OA and other joint defects. At 10 weeks after an allogeneic adipose-derived MSCs injection, the dogs exhibited improved symptoms including pain reduction, improvement of mobility, increased daily activity, and increased quality of life. The study [66] revealed that the intra-articular injection of MSCs achieved better effects than intravenous injection. Moreover, according to Cabon et al. [67], delivery of single or multiple intra-articular injections of allogeneic MSCs demonstrated significant clinical improvements in dogs with refractory osteoarthritis for a period of 2 years. Kriston-Pál et al. [68] transplanted adipose tissue derived MSCs to 58 dogs with OA and followed up in 4–5 years. The results showed that 83% of these dogs improved or retained improvement in lameness, and no serious adverse event or treatment-related death was observed. Vilar et al. [69], however, stated that the effect of MSCs therapy can only last for 3 months in dogs with hip OA. We assume the difference may have resulted from differences between the breeds or techniques involved, and it is worth noting that these studies are not randomized controlled trials. 

Randomized controlled clinical trials conducted in dogs provide evidence for the effectiveness of stem cell treatment for OA. Black et al. found that dogs suffering from chronic OA of the coxofemoral joints showed significant improvement in their gait and a decrease in lameness after receiving intra-articular injections of adipose-derived stem cells compared to the control group [70]. Cuervo et al. [71] demonstrated that a single intra-articular injection of adipose-derived MSCs is safe and effective in dogs with hip OA, and results in a reduction in pain and an improvement in physical function at 6 months post-treatment. Notably, this treatment showed superior efficacy compared to an injection of plasma rich in growth factors. Harman et al. [72] and Maki et al. [73] reached a similar conclusion, showing that allogeneic adipose MSCs are efficacious in canine OA compared to placebo. In addition, Kim et al. [74] conducted a double-blinded, placebo-controlled clinical trial and found that intra-articular umbilical cord-derived MSCs improved clinical signs in dogs with elbow OA at 6 months with no adverse effects. Together, these findings provide strong evidence for the potential of stem cell therapy as a safe and effective treatment option for OA in dogs. 

In addition to joint functional assessment, studies in osteoarthritic dogs have reported improvements in inflammation biomarkers and cartilage evaluation. Maki et al. [73] reported that after MSCs treatment, serum interleukin 10 was increased in most of the dogs with hip OA as a potential biomarker for improved lameness. Zhang et al. [75] applied umbilical cord-derived MSCs in an osteoarthritic canine model and found significantly lower blood levels of IL-6, IL-7, and TNF-α in the treated group compared to the untreated control group [75], demonstrating reduced inflammation. Scanning electron microscope results showed that the cartilage neogenesis in the treated group had more visible neonatal tissue and new tissue fibers than those in the control group [75], indicating significantly improved cartilage neogenesis and recovery. Li et al. [76] treated osteoarthritic dogs with a combination of bone marrow MSCs, and hyaluronic acid. At 28 weeks after treatment, magnetic resonance imaging (MRI) and histological assessment showed that the combined treatment resulted in cartilage-like tissue regeneration and improvement in cartilage defects compared to both hyaluronic acid alone and the control group [76]. However, these results were obtained from animal disease models rather than natural diseases in clinical patients, and as such the applicability of the research findings is questionable. A study [77] conducted on dogs with naturally occurring elbow OA revealed that after a year of treatment with allogeneic MSCs dispersed in a hyaluronic acid solution, the cartilage had regenerated and histological analysis of the cartilage biopsy by arthroscopy indicated that the regenerated cartilage was of the hyaline type, further supporting the effect of stem cell therapy in OA [77]. 

Hip dysplasia (HD), characterized by instability and luxation or subluxation of the hip, is an inherited orthopedic pathology that affects dogs, causing pain and lameness. Elbow dysplasia (ED), a condition involving multiple developmental abnormalities of the elbow, may cause front limb lameness in dogs. HD and ED are common causes of OA in these joints. Although genetic factors play a pivotal role in the development of HD and ED [78], gene therapy is not yet an established treatment option. The current treatments for dysplasia include conservative measures such as exercise restriction and the administration of analgesics or chondroprotective agents, as well as surgical procedures. However, these treatments often have limited efficacy and can be invasive. Stem cell therapy has emerged as a promising alternative for the treatment of HD and ED. Kriston-Pal et al.’s study [77] treated osteoarthritic dogs with ED with adipose tissue-derived MSCs and demonstrated a significant improvement in the degree of lameness at the 1-year follow-up. Marx et al. [79] conducted a study on dogs suffering from HD who exhibited a weak response to conventional drug therapy. Adipose-derived stem cells were injected in three acupuncture points near the affected joint, and data showed that the acupoint injection of stem cells resulted in improvement in the range of motion, lameness at trot, and joint pain in the treated dogs [79].

Cranial cruciate ligament (CCL) rupture is another important veterinary health problem in dogs [53,80]. It is characterized by degeneration of the extracellular matrix of the ligament which eventually leads to ligament rupture and is primarily relevant to a dog’s body weight and spay/neuter status [53]. In surgical treatments of CCL rupture in dogs, MSCs showed effects in promoting bone regeneration and healing. Taroni et al. [81] compared the therapeutic effects of a single intra-articular injection of allogeneic MSCs and a 1-month course of non-steroidal anti-inflammatory drugs (NSAIDs) in dogs that underwent tibial plateau leveling osteotomy due to CCL rupture. At 1 month postoperatively, MSCs significantly boosted bone healing, although there was no long-term difference observed between the two groups in terms of clinical scores and gait evaluation [81]. Similarly, Santos et al. [82] reported a 30-day positive effect of MSCs after tibial tuberosity advancement in dogs with CCL rupture, but longer-term effects were doubtable. Linon et al. [83] treated dogs with partial or complete CCL rupture with autologous bone marrow cell implantation in addition to tibial plateau levelling osteotomy. Three months after treatment, 7 out of 8 dogs exhibited reduced lameness and pain, as well as improved articular function. At 1 year after treatment, all dogs had resumed normal activities [83]. Moreover, some studies have shown that dogs with CCL rupture who do not undergo surgery can still achieve satisfactory long-term outcomes. A retrospective study [84] evaluated dogs with a partial CCL tear treated by stem cell therapy or BMAC, combined with PRP, and demonstrated that after treatment the treated dogs gained mobility and stifle arthroscopy showed intact and normal CCL in most dogs. In this study, the exact stem numbers injected were not clear, and no conclusion indicated whether “pure” stem cell or BMAC is superior [84]. Muir et al. [80] demonstrated that the use of autologous bone marrow-derived MSCs in dogs with CCL rupture resulted in reduced systemic and local inflammation, as evidenced by lower levels of serum C-reactive protein (CRP), synovial CRP, and synovial interferon-gamma (IFN-γ).

Moreover, the use of MSCs has also shown promising results in the treatment of tendon injuries. In Canapp et al.’s study [85], dogs aged 1–14 years with supraspinatus tendinopathy were treated with adipose-derived progenitor cells and platelet-rich plasma. Cells and plasma were injected into the tendon lesion under ultrasound guidance. At 90 days following injection, the dogs showed improvements in objective gait analysis and pathology rating scale. Case et al. [86] published a clinical report that a dog with gastrocnemius tendon strain was treated with autologous MSCs and orthosis. Lameness was alleviated and normal linear fiber pattern was partly restored after treatment.

## 5. Adverse Events

In the above-mentioned studies, only minor side effects were reported, such as mild pain [71], nausea and vomiting [49,51], slight discomfort [66], mild skin allergy [66], short-term joint swelling [68], self-limiting inflammatory joint reactions or local inflammation [67,68], and transitory worsening of lameness related to incorrect technique [69]. In Daems et al.’s study [87], a dog was reported to have diarrhea and vomiting for 4 days at 1 week after an intra-articular administration of equine chondrogenic-induced MSCs; however, this was presumably due to the concomitant NSAID administration and thus unlikely to be related to the stem cell therapy. 

It should also be noted that a few studies reported more severe complications in animals who underwent stem cell therapy. Kang et al. [88] injected allogeneic bone- marrow-derived MSCs into healthy dogs in a laboratory setting and considered the resulting pulmonary edema and hemorrhage to be possible adverse reactions. Vulliet et al. [89] reported that intra-coronary arterial injection of MSCs in healthy dogs caused acute myocardial ischemia and subacute myocardial microinfarction, indicating that the approach of cell administration is important. Quimby et al. [49] reported that a cat developed overt respiratory distress after multiple infusions of MSCs, although the cat recovered after receiving prompt medical attention in a critical care unit. 

Graft-versus-Host Disease (GVHD) is also possible in dogs. Schaefer et al. [90] reported a case in which a dog underwent reduced-intensity hematopoietic stem cell transplantation from a leukocyte antigen-identical littermate, resulting in GVHD that ultimately led to euthanasia at day 52 [90]. These adverse events may be associated with unintended immune reactions and should be carefully observed and monitored in future studies.

## 6. Challenges

Stem cells have been shown to be safe and effective to use in the treatment of aging and joint diseases (as summarized in Table 1), largely due to their anti-inflammatory properties and immune modulation capabilities [27]. However, the use of stem cells in veterinary medicine is still in its early stages, and many challenges and limitations need to be addressed before stem cell therapy can become a widely accepted treatment option for pets. One of the major challenges is that current trials of stem cell therapies in companion animals lack long-term safety and efficacy data, which poses difficulties in fully evaluating the benefits and risks. In addition, there are no consensus guidelines or regulations governing the use of stem cells in pets at present [91]. This makes it difficult to ensure that the procedures are performed safely and effectively and as well as ethical considerations. The standard of stem cell therapy also needs to be explored. For example, studies discussed in this review paper showed significant variations in the doses of stem cells used (shown in Table 1), which may result in a potential discrepancy in therapeutic efficacy. Furthermore, there is a need for more research to better understand the mechanisms by which stem cells exert their therapeutic effects in different disease conditions and to identify the optimal sources and types of stem cells for different applications. 

## 7. Conclusions

Despite the challenges and limitations of stem cell therapy, it has been found to be safe and effective in treating aging-related and orthopedic diseases in pets with minimal side effects. Stem cell therapy has the potential to relieve symptoms and improve the quality of life of our animal companions, for their benefit as well as ours. Although further research is needed to fully understand the potential of stem cells in veterinary medicine, current evidence supports their use as a valuable tool in managing many conditions. Continued research in this field holds promise for further advancing the health and well-being of our animal friends.

## Figures and Tables

**Table 1 animals-13-02457-t001:** Summary of clinical reports.

Therapeutic Indication	Authors	Patients	Stem Cell Therapy	Control Group	Follow-Up Duration	Major Outcome Measures	Key Efficacy Results of Stem Cell Therapy
Cognitive Dysfunction Syndrome	Valenzuela et al. [42]	6 dogs, 10–16 years old	Autologous skin-derived neural precursor cells, 0.25 × 10^6^ cells, bilateral hippocampus injection	No treatment (media-only)	3 months	Canine Cognitive Dysfunction Rating Scale (CCDR), pathology	Improvement in CCDR, increase in synaptic makers
Congestive heart failure secondary to myxomatous mitral valvular disease	Yang et al. [43]	10 dogs, 12.8 ± 0.8 years old	Allogeneic wharton jelly derived mesenchymal stem cells (MSCs) in a 1% solution of autologous serum, 2 × 10^6^ cells/kg, three intravenous injections	Autologous serum	6 months	Echocardiograph, electrocardiogram (ECG), serum cardiac biomarkers, complete blood count (CBC), serum biochemical analysis, survival time, and time to first diuretic drug dosage escalation	Decreases in lymphocyte, eosinophil, and monocytes; no significant differences between groups in echocardiographic variables, ECG results, serum cardiac biomarker concentrations, survival time, and time to first diuretic drug dosage escalation
Myxomatous mitral valvular disease	Petchdee and Sompeewong [45]	20 dogs, 8–15 years old	Allogeneic puppy deciduous teeth stem cells, 1 × 10^6^ cells/kg, two intravenous injections	Standard treatment and PBS injection	2 months	Electrocardiography, complete transthoracic echocardiography, thoracic radiography, and blood pressure	Improvements in left ventricular ejection fraction, American College of Veterinary Internal Medicine functional class, and quality of life
Chronic kidney disease	Thomson et al. [47]	5 cats, 8.5–17 years old	Autologous adipose tissue derived MSCs, 1.5–6 × 10^6^ cells, two intra-arterial injections to the kidney	None	3 months	CBC, serum biochemistry, urinalysis, urine culture, urine protein: creatinine (UPC) ratio, blood pressure, iohexol clearance, and evaluation forms	Improvements in overall condition, decrease in polydipsia, improvements in energy and activity levels
	Quimby et al. [48]	10 cats, 6–17 years old	Autologous adipose tissue-derived MSCs, bone marrow-derived MSCs, different doses (1 × 10^5^ cells, or 1 × 10^6^ cells, or 2 × 10^6^ cells, or 4 × 10^6^ cells), unilateral intrarenal injection into kidney	2 young healthy cats	Up to 4 months	Glomerular filtration rate (GFR), CBC, serum biochemistry, urinalysis, UPC ratio, and histopathology	Increase in GFR and decrease in serum creatinine in some animals
	Quimby et al. [49]	21 cats, 7–18 years old	Allogeneic adipose tissue derived MSCs, 2 × 10^6^ cells or 4 × 10^6^ cells, three intravenous injections	No treatment	4–8 weeks	Serum biochemistry, CBC, urinalysis, urine protein, GFR, and urinary cytokine	Decrease in serum creatinine concentrations in some animals, but not clinically relevant
	Quimby et al. [50]	8 cats, 9–15 years old	Allogeneic adipose tissue derived MSCs, 2 × 10^6^ cells/kg, three intravenous injections	Placebo	8 weeks	Serum biochemistry, complete blood count, urinalysis, urine protein, GFR, and UPC ratio	No significant change in serum creatinine, blood urea nitrogen, potassium, phosphorus, GFR, UPC ratio, or packed cell volume
	Vidane et al. [51]	9 cats, 8–16 years old	Allogeneic amniotic membrane derived MSCs, 2 × 10^6^ cells, two intravenous injections	1 healthy cat	2 months	CBC, serum biochemistry parameters, urinalysis, UPC, blood gases and electrolytes	Decreases in UPC ratio and serum creatinine concentration, increases in food intake and social behavior
Osteoarthritis	Wits et al. [26]	12 dogs, 0.6–5 years old	Allogeneic adipose- tissue-derived MSCs with or without hyaluronic acid, 5 × 10^6^ cells, intra-articular injection	Placebo	3 months	Lameness, pain on manipulation, articular edema, range of motion, muscle atrophy, detection of crepitus on hip rotation and abduction, radiographic examination, Norberg angle measurements	Decreases in lameness at walk and pain on manipulation, improvements in range of motion and detection of crepitus on hip rotation and abduction
	Okamoto-Okubo [27]	16 dogs, aged 6.3 ± 3.9 and 4.6 ± 2.3 years in stem cell group and the other group	Allogeneic adipose- tissue-derived MSCs, 18 × 10^6^ cells, two intra-articular injections	Platelet-rich plasma	2 months	Canine Brief Pain Inventory (CBPI), the Helsinki Chronic Pain Index (HCPI), Visual Analogue Scales (VAS) for pain and locomotion, force plate gait analysis, response to palpation, descriptive numerical scale for pain	Decreases in HCPI, CBPI, VAS-pain, VAS-palp scores, as well as pain interference score and pain severity score, at 2 months
	Sanghani-Kerai [28]	25 dogs, 1.5–11.5 years old	Autologous adipose tissue-derived MSCs and platelet-rich plasma, intra-articular injection	None	6 months	Liverpool Osteoarthritis in Dogs (LOAD) score, Modified Canine Osteoarthritis Staging Tool (mCOAST), kinetic gait analysis, and diagnostic imaging	Decreases in LOAD score and Asymmmetry indices, improvements in mCOAST and quality of life
	Shah et al. [66]	203 dogs, 0.7–16 years old	Allogeneic adipose- tissue-derived MSCs, single intra-articular injection and/or single intravenous injection	None	10 weeks	Symptoms (lameness and pain scores) and quality of life	Improvements in pain, mobility, daily activity, and quality of life
	Cabon et al. [67]	22 dogs, 1–10 years old	Allogeneic neonatal tissues derived MSCs, 2 × 10^6^ cells or more, 1 or 2 intra-articular injections	None	2 years	Clinical symptoms and questionnaires	Clinical benefits
	Kriston-Pál et al. [68]	58 dogs, 0.4–10 years old	Allogeneic adipose tissue derived MSCs, 12 × 10^6^ ± 3.2 × 10^6^ cells, intra-articular injection	None	Up to 5 years	Questionnaire (including degree of lameness)	Improvement in lameness
	Vilar et al. [69]	10 dogs, 4–8 years old	Autologous adipose tissue derived MSCs, intra-articular injection	5 healthy dogs	6 months	Gait analysis (peak vertical force and vertical impulse)	Increases in peak vertical force and vertical impulse, but only within the first 3 months
	Black et al. [70]	21 dogs, 1–11 years old	Autologous adipose-derived stem cells, 4.2 × 10^6^ cells, intra-articular injection	Placebo	3 months	Scores for lameness, pain, range of motion, and functional disability	Improvements in lameness, pain, and range of motion
	Cuervo et al. [71]	39 dogs, 0.7–11.3 years old	Autologous adipose-derived MSCs, 30 × 10^6^ cells, intra-articular injection	Plasma rich in growth factors	6 months	Pain Assessment (VAS), degree of osteoarthritis, Bioarth scale assessment, quality of life	Improvements in function, pain, range of motion, and quality of life
	Harman et al. [72]	74 dogs, aged 7.98 ± 3.56 and 8.59 ± 3.53 years in treatment and control groups	Allogeneic adipose-derived MSCs, 12 × 10^6^ cells, intra-articular injection	Placebo	2 months	Client-specific outcome measurement scoring, pain on manipulation, veterinary and owner assessment of clinical outcomes	Improvements in client-specific outcome measurement scoring, pain on manipulation, and veterinary global score
	Maki et al. [73]	20 dogs, 1–14 years old	Allogeneic adipose-derived MSCs, different doses (5 × 10^6^, or 25 × 10^6^, or 50 × 10^6^ cells), intra-articular injection	Placebo	3 months	Lameness and pain; serum levels of interleukin-1 receptor antagonist protein and interleukin-10	Improvement in lameness scores and pain in all doses; increase in serum interleukin-10
	Kim et al. [74]	55 dogs, 1–11 years old	Allogeneic umbilical- cord-derived MSCs, 7 × 10^6^ cells, intra-articular injection	Placebo	6 months	Force-platform gait analysis, Hudson Visual Analog Scale (HVAS)	Improvements in CBPI and HVAS scores
	Kriston-Pal et al. [77]	30 dogs	Allogeneic adipose- tissue-derived MSCs in hyaluronic acid, 12 × 10^6^ ± 3.2 × 10^6^ cells, intra-articular injection	None	1 year	Questionnaire (lameness), arthroscopy, histological analysis	Improvement in lameness, increase in cartilage regeneration
	Daems et al. [87]	6 dogs, 5–10 years old	Xenogeneic chondrogenic induced MSCs, 1 × 10^6^ cells, intra-articular injection	Placebo	12 weeks	Pressure plate analysis, orthopedic examination, synovial fluid analysis, radiographic examination, owner surveys	Improvements in pain and lameness; no significant differences in the orthopedic examination parameters, the radiographic examination, synovial fluid sampling, and pressure plate analysis between groups
Elbow dysplasia	Kriston-Pal et al. [77]	30 dogs	Allogeneic adipose- tissue-derived MSCs, 12 × 10^6^ ± 3.2 × 10^6^ cells, in hyaluronic acid, intra-articular injection	None	1 year	Questionnaire (lameness), arthroscopy, histological analysis	Improvement in lameness, increase in cartilage regeneration
Hip dysplasia	Marx et al. [79]	9 dogs, 0.5–12 years old	Autologous adipose-derived stem cells, 2–8 × 10^5^ cells, or vascular stromal fractions, 2–5 × 10^6^ cells, acupoint injection	None	1 month	Range of motion, lameness at trot, pain on manipulation	Improvements in range of motion and lameness, decrease in pain on manipulation
Cranial cruciate ligament rupture	Muir et al. [80]	12 dogs, 1.6–9.6 years old	Autologous bone- marrow-derived MSCs, 2 × 10^6^ cells intravenous and 5 × 10^6^ cells intra-articular injection, after tibial plateau leveling osteotomy	None	8 weeks	Circulating T lymphocyte subsets, C-reactive protein (CRP) and cytokine concentrations in serum and synovial fluid, total nucleated cell count in synovial fluid, radiography	Decrease in serum CRP, synovial CRP, synovial interferon-gamma-Υ; increase in serum chemokine ligand 8
	Taroni et al. [81]	14 dogs, 1–13 years old	Allogeneic neonatal tissues derived MSCs, 10 × 10^6^ cells or more, intra-articular injection, after tibial plateau leveling osteotomy	Standard treatment	6 months	Clinical score, bone healing radiographic score, semiquantitative gait evaluation	Increase in bone healing scores at 1 month; no significant difference in clinical scores and gait evaluation between the two groups at 1, 3, 6 months
	Santos et al. [82]	9 dogs, 1–12 years old	Autologous adipose- tissue-derived MSCs, 1.5 × 10^6^ cells, percutaneous injection at the osteotomy site, after tibial tuberosity advancement	No treatment (media-only)	4 months	Radiographs (scores and analysis of the density of bone trabeculae in the spongy substance)	Increase in ossification at 1 month, but no difference for the other period
	Linon et al. [83]	7 dogs	Autologous bone marrow cells, 1 × 10^7^ cells, intra-articular injection, before tibial plateau levelling osteotomy	None	1 year	Fluorescence microscopy of synovial membrane, synovial fluid analyses, clinical assessment	Increase in numbers of engrafted cells; improvements in lameness, function, activity, and pain on manipulation
	Canapp et al. [84]	36 dogs, 1–9.5 years old	Autologous bone marrow aspirate concentrate or adipose tissue-derived progenitor cells, combined with platelet-rich plasma, intra-articular injection	None	3 months	Orthopedic evaluation, radiograph, objective gait analysis, diagnostic stifle arthroscopy, and functional questionnaire	Improvements in stifle arthroscopy findings, total pressure index percent, and quality of life; surgery needed in some animals
Supraspinatus tendinopathy	Canapp et al. [85]	55 dogs aged 1–14 years	Autologous adipose tissue-derived progenitor cells, 5 × 10^6^ cells, combined with platelet-rich plasma, injected into the tendon lesion	None	3 months	Orthopedic evaluation, radiograph, magnetic resonance imaging, musculoskeletal ultrasonography, objective gait analysis, and arthroscopy	Increase in total pressure index percentage, reduction in tendon size, improvement in fiber patterns
Gastrocnemius tendon strain	Case et al. [86]	1 dog, 4 years old	Autologous MSCs and orthosis, >20 × 10^6^ cells, injected into the tendon core lesion	None	631 days	Serial orthopedic examinations, ultrasonography, force-plate gait analysis	Improvement in lameness, increase in peak vertical force, improvement in fiber pattern

## Data Availability

No new data were created or analyzed in this study. Data sharing is not applicable to this article.

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
