# Peer review of "Stem Cell Therapy for Aging Related Diseases and Joint Diseases in Companion Animals"

_animals, 2023, doi:10.3390/ani13152457_

Round 1

Reviewer 1 Report

The subject of the review is interesting and very topical, I congratulate you on the idea.

Literature reviews are always a complex exercise because they must summarize the knowledge acquired over a long period of time while maintaining a critical view of what has been written.

Personally I find this review to be significantly improved, the treatment at the moment is very simplistic and in some points it also expresses incorrect concepts.

I have included only a few comments in the text, to give an idea of what I mean.

Furthermore, no mention is made of the limitations that some studies reported in the literature report, for example in some cases the use of only clinical and therefore non-objective evaluations, the small number of cases, the absence of control groups, etc.

The authors reported the positive aspects of treatment with stem cells, but without transferring to the reader the right amount of critical sense that he must have in studying the literature.

Therefore I invite the authors to significantly improve the text and resubmit it .

Reviewer 2 Report

Thank you the opportunity to review your manuscript “Stem cell therapy for aging and orthopedics in the pet industry”. The manuscript provides a review of types of stem cells and reported applications for age-related and orthopedic diseases. The manuscript is a good attempt at a review of what is reported in the literature for use of stem cells in companion animals for age-related and orthopedic diseases. In addition to the line-by-line comments provided, I have several suggestions for expanding this review to provide the reader a more thorough understanding and overview of the literature, as I feel the review as written may be overly simple and brief. I would also urge the authors to comment on how this review fills a gap of knowledge compared to other stem cell reviews already in publication. 

  • An introduction to the definition of autogenous vs allogeneic stem cells (+/- xenogeneic), pros and cons, unique properties of stem cells that make allogeneic use possible 

  •  
  • Further commentary on the different sources for MSCs, such as common sites for bone marrow sampling or adipose tissue collection, pros and cons associated with each 

  •  
  • Also consider discussing stromal vascular fraction and BMAC, how these compare to pure adipose- or bone marrow-derived stem cell products 

  •  
  • A brief, general explanation of how tissues are handled following collection and isolated/cultured/expanded into stem cell products 

  •  
  • Relating to specific studies, doses of stem cells tend to be highly variable. Either including number of stem cells for each study mentioned (when available) in the text or in a table format, and including a discussion of the possible discrepancy in stem cell efficacy when used therapeutically due to variation in dosing

  •  

  • Additionally, various scaffolds such as platelet products or hyaluronic acid may be used in conjunction with stem cells in these studies and generally there appears to be a benefit of co-administration of PRP and stem cells. Please discuss. 

  •  
  • In general, a table for each major therapeutic indication you discuss with the papers and key points from the paper (number of animals, whether control group was used, major outcome measures, etc.) would be very helpful (e.g. separate tables for OA, soft tissue, and other age-related diseases) 

  •  
  • A quick search identified the following additional recently published articles that could be considered as highly relevant to your review 

  •   
  • Pavarotti GS, Hivernaud V, Brincin M, Roche R, Barreau P, Festy F, et al.. Evaluation of a single intra-articular injection of autologous adipose tissue for the treatment of osteoarthritis: a prospective clinical study in dogs. Vet Comp Orthop Traumatol. (2020) 33:258–66 

  •  
  • Cabon Q, Febre M, Gomez N, Cachon T, Pillard P, Carozzo C, et al.. Long-term safety and efficacy of single or repeated intra-articular injection of allogeneic neonatal mesenchymal stromal cells for managing pain and lameness in moderate to severe canine osteoarthritis without anti-inflammatory pharmacological support: pi. Front Vet Sci. (2019) 6:e10 

  •  
  • Maki CB, Beck A, Wallis CBCC, Choo J, Ramos T, Tong R, et al.. Intra-articular administration of allogeneic adipose derived mscs reduces pain and lameness in dogs with hip osteoarthritis: a double blinded, randomized, placebo controlled pilot study. Front Vet Sci. (2020) 7:e570 

  •  
  • Skangals U, Ilgaz?s A. Stem cell therapy in the treatment of bilateral elbow joint osteoarthritis in dogRes Rural Dev. (2019) 1:252–7 

  •  
  • Hunáková K, Hluchý M, Špaková T, Matejová J, Mudronová D, Kuricová M, et al.. Study of bilateral elbow joint osteoarthritis treatment using conditioned medium from allogeneic adipose tissue-derived MSCs in Labrador retrievers. Res Vet Sci. (2020) 132:513–20 

  •  
  • Kriston-Pál É, Haracska L, Cooper P, Kiss-Tóth E, Szukacsov V, Monostori É, et al.. Regenerative approach to canine osteoarthritis using allogeneic, adipose-derived mesenchymal stem cells. Safety results of a long-term follow-up. Front Vet Sci. (2020) 7:e510 

  •  
  • Wits MI, Tobin GC, Silveira MD, Baja KG, Braga LMM, Sesterheim P, et al.. Combining canine mesenchymal stromal cells and hyaluronic acid for cartilage repair. Genet Mol Biol. (2020) 43 

  •  
  • Okamoto-Okubo CE, Cassu RN, Joaquim JGF, Dos Reis Mesquita L, Rahal SC, Oliveira HSS, et al.. Chronic pain and gait analysis in dogs with degenerative hip joint disease treated with repeated intra-articular injections of platelet-rich plasma or allogeneic adipose-derived stem cells. J Vet Med Sci. (2021) 83:881–8 

  •  
  • Sanghani-Kerai A, Black C, Cheng SO, Collins L, Schneider N, Blunn G, et al.. Clinical outcomes following intraarticular injection of autologous adipose-derived mesenchymal stem cells for the treatment of osteoarthritis in dogs characterized by weight-bearing asymmetry. Bone Jt Res. (2021) 10:650–8 

Specific Comments: 

Line 17-18: “...ailments that are not curable but conventional veterinary medicine.” Please correct typo “but” in this sentence 

Line 30: Consider additional keywords “regenerative medicine”, “aging” 

Line 61: “...animal companions.” Please maintain consistency in using “companion animals” described the population of animals the manuscript will focus on. 

Line 80-81: “...although site-specific differences have been observed regarding stem cell numbers and differentiation capacity.” Can you briefly expand on the major differences here? 

Line 87-88: Awkward sentence here, please edit “As a result, MCSs have been shown efficacy...” 

Line 121: “...allogeneic Wharton jelly-derived...” Can you please provide clarification for readers that this is an umbilical cord tissue. 

Line 136: Awkward wording here, please edit “...demonstrated the safety of feasibility of this treatment.” 

Line 150-152: Can you please clarify in this paragraph if there are any studies of the use of stem cells for these additional age-related conditions? For example - “...for which the use of stem cell therapy is currently not widely practiced in the veterinary industry and for which no clinical studies are available.” or something of that nature. 

Line 156: Awkward wording here, please edit “...but these treatments are not always be effective...” 

Line 162-176: Consider separating the discussion of dog and cat OA into separate paragraphs as the transition from one to the other was rather jarring. I think two separate paragraphs would also better highlight the differences in OA in dogs versus cats. 

Line 174: Referring to reference 57, what age range of cats was used in this study population? The reported prevalence of OA in cats varies widely in the literature depending on age of the study population; including this information for reference 57 in your text would be helpful to the reader. 

Line 181: “...and or...” should be “...and/or...” 

Line 192-193: It would be helpful to provide clarification if the studies discussed in the preceding paragraph (Shah et al, Cabon et al, Vilar et al) are not randomized controlled trials, otherwise it may be confusing to the reader why these studies have been separated from those that are about to be discussed in the following paragraph. 

Line 281: “...in the real world...” may be more appropriately replaced with “...in clinical patients...” 

Line 219: Superscript reference 68 should read [68]

Line 236: Referring to reference 70, please provide clarification of where the stem cell injection was performed (e.g. local acupoints surrounding the coxofemoral joint) 

Line 261-266: This paragraph stands out as it is the only portion of the manuscript where you talk about horses. There are other horse OA papers that you do not discuss. I understand the desire to include these studies here as there is so little on the small animal side, but either horses should be included throughout the paper or it needs to be very clear why you are discussing horses in this section and not elsewhere (e.g. due to the extreme lack of comparable studies on the small animal side). If you do wish to leave horses included in this section, it should be more comprehensive (there are at least a few other studies in equines using stem cells for tendon injuries). Consider including the following references which discuss the use of stem cells specifically in canine patients for musculotendinous injuries such as the following:

  • Canapp SO, Canapp DA, Ibrahim V, Carr BJ, Cox C, Barrett JG. The use of adipose-derived progenitor cells and platelet-rich plasma combination for the treatment of supraspinatus tendinopathy in 55 dogs: a retrospective study. Front Vet Sci. 2016;3(61) 

  •  

Line 279 and 283: For these two references (79 and 80), how were stem cells administered? Is it possible that route of administration affected these adverse reactions (especially for the healthy dogs who developed pulmonary edema and hemorrhage)?

Reviewer 3 Report

the review By Wang and coauthors is well written, clear, amd covers adequately the  implications concerning the use of cell terapies in veterinary medicine. I only suggest to change the title into 'Stem Cell Therapy for age related diseases and Orthopedics in companion animals'.

The main question addressed by this research is if the use of MSC appropriate to set up therapeutic protocols for the treatment of age related diseases and osteoarthritis in companion animals

Since the manuscript is a review, we cannot consider it like an experimental investigation that must addresses a specific gap in the field. I think that the review covers properly all the aspects of the topic approached

The conclusions are consistent with the evidence and arguments presented and they address the main question.

The references are appropriate

I think that this review doesn’t need to report figures or tables

Round 2

Reviewer 1 Report

The authors have improved the manuscript and I thank them.

However, this literature review remains limited and not exhaustive. It should deal with the use of Stem Cell Therapy for Aging Related Diseases and Orthopedics in Companion Animals, but in the orthopedics session, for example, it deals with the use of stem cells in some joint pathologies (hip, elbow and knee) and then just nods of 4 lines on tendinopathies. I suggest either treating better orthopedics disease or just treating joint disease at this point.

Author Response

Dear reviewers,

Thank you for your review and for your valuable comments. We have responded to your comments and revised our manuscript accordingly.

Reviewer 1

This literature review remains limited and not exhaustive. It should deal with the use of Stem Cell Therapy for Aging Related Diseases and Orthopedics in Companion Animals, but in the orthopedics session, for example, it deals with the use of stem cells in some joint pathologies (hip, elbow and knee) and then just nods of 4 lines on tendinopathies. I suggest either treating better orthopedics disease or just treating joint disease at this point.

R: Agreed. We have now changed our title and subtitles from “orthopedics disease” to “joint disease”. Thank you for your comments.

Reviewer 2 Report

Thank you for the opportunity to review your revised manuscript, your time into improving this manuscript is appreciated and I think the revisions improve the potential impact of this paper. There are a few areas where I think some edits would improve clarity and flow of the paper. 

Line 20: Please consider clarifying in your simple summary that this review will summarize uses of stem cells as treatment of age-related and orthopedic diseases in companion animals. It is clear in the title and abstract, would be nice for it to be clearly stated in the simple summary as well. 

Line 81-84, 87-90, 99-114: Thank you for these additions! 

Line 190-191 and Line 197-198: “Around one in four of the 77.2 million pet dogs in the US are diagnosed with some form of arthritis.” “In fact, data suggests that OA affects more than 20% of dogs in the US who are over the age of one.” These sentences pretty much say the same thing from the same reference, can they be combined into one sentence? 

Line 199: This sentence is a little confusing; as written, it seems like it is talking about two distinct subjects (pain scores in dogs are easily interpreted, OA in cats is underdiagnosed). Would you please edit to make the flow of this sentence smoother and improve reading comprehension? I think making it clearer that because pain secondary to OA is more easily identified in dogs than in cats, suspicion of and subsequent diagnosis of OA is therefore more common in dogs than in cats. 

Line 260-276: I’m a little confused about the organization of this paragraph and the preceding paragraphs. HD and ED are the most common cause of OA in these joints, were the Kriston-Pal et al and Marx et al studies treating dogs with HD/ED prior to radiographic signs of OA and therefore targeting other aspects of clinical disease? Otherwise, I am confused why these are separated out from say, the Kim et al study which also looked at dogs with elbow OA of which most if not all were likely secondary to ED. 

Line 312-337: Consider pulling these sections out into a separate “Adverse Effects” section rather than including in the Discussion section. 

Table 1: Thank you for adding this table! I think it serves as a great reference to extract important information to accompany the text. Would it be possible to add a column to briefly describe key results of each study? This would allow the reader a helpful glimpse into bigger picture findings (such as contradictory results of the MMVD studies but the variation into the study design which may explain the contradiction). 

Also Table 1: Number of dogs not listed for the Canapp et al paper. 

Author Response

Dear reviewers,

Thank you for your review and for your valuable comments. We have responded to your comments and revised our manuscript accordingly.

Reviewer 2

  1. Line 20: Please consider clarifying in your simple summary that this review will summarize uses of stem cells as treatment of age-related and orthopedic diseases in companion animals. It is clear in the title and abstract, would be nice for it to be clearly stated in the simple summary as well. 

R: It has been clarified.

  1. Line 190-191 and Line 197-198: “Around one in four of the 77.2 million pet dogs in the US are diagnosed with some form of arthritis.” “In fact, data suggests that OA affects more than 20% of dogs in the US who are over the age of one.” These sentences pretty much say the same thing from the same reference, can they be combined into one sentence? 

R: These two sentences have been combined into one: “Around one in four of the 77.2 million pet dogs in the US are diagnosed with some form of arthritis, with over 20% of them being affected by OA once they are over the age of one.”

  1. Line 199: This sentence is a little confusing; as written, it seems like it is talking about two distinct subjects (pain scores in dogs are easily interpreted, OA in cats is underdiagnosed). Would you please edit to make the flow of this sentence smoother and improve reading comprehension? I think making it clearer that because pain secondary to OA is more easily identified in dogs than in cats, suspicion of and subsequent diagnosis of OA is therefore more common in dogs than in cats. 

R: The sentence has been rephrased: “Suspicion of and subsequent diagnosis of OA is more difficult in cats than in dogs, as pain secondary to OA is more easily interpreted in dogs, and cats tend to use behavioral strategies to disguise pain and lameness.”

  1. Line 260-276: I’m a little confused about the organization of this paragraph and the preceding paragraphs. HD and ED are the most common cause of OA in these joints, were the Kriston-Pal et al and Marx et al studies treating dogs with HD/ED prior to radiographic signs of OA and therefore targeting other aspects of clinical disease? Otherwise, I am confused why these are separated out from say, the Kim et al study which also looked at dogs with elbow OA of which most if not all were likely secondary to ED. 

R: Marx et al. (ref #79) studied dogs with HD rather than dragonized OA, so it was discussed separately. In Kriston-Pal et al.’s study (ref #77), the dogs had ED-associated OA, and both ED and OA were evaluated. On the other hand, the study conducted by Kim et al. (ref #74) focused on dogs with elbow OA, but there was no mention of elbow ED in the dogs. Therefore, we have separated it into a different section. A statement has been added to the paragraph, noting that HD and ED are common causes of OA in these joints.

  1. Line 312-337: Consider pulling these sections out into a separate “Adverse Effects” section rather than including in the Discussion section. 

R: Agreed. These paragraphs have been separated into the Adverse Effects section.

  1. Table 1: Thank you for adding this table! I think it serves as a great reference to extract important information to accompany the text. Would it be possible to add a column to briefly describe key results of each study? This would allow the reader a helpful glimpse into bigger picture findings (such as contradictory results of the MMVD studies but the variation into the study design which may explain the contradiction). 

R: A column describing key results has been added.

  1. Also Table 1: Number of dogs not listed for the Canapp et al paper. 

R: The number of dogs has been added. Thank you for your review.